# Practical Camera Sensor Spectral Response and Uncertainty Estimation

**DOI:** 10.3390/jimaging6080079

**Published:** 2020-08-05

**Authors:** Mikko E. Toivonen, Arto Klami

**Affiliations:** Department of Computer Science, University of Helsinki, 00560 Helsinki, Finland; arto.klami@helsinki.fi

**Keywords:** spectral response, calibration, stochastic optimization, data fusion, uncertainty estimation

## Abstract

Knowledge of the spectral response of a camera is important in many applications such as illumination estimation, spectrum estimation in multi-spectral camera systems, and color consistency correction for computer vision. We present a practical method for estimating the camera sensor spectral response and uncertainty, consisting of an imaging method and an algorithm. We use only 15 images (four diffraction images and 11 images of color patches of known spectra to obtain high-resolution spectral response estimates) and obtain uncertainty estimates by training an ensemble of response estimation models. The algorithm does not assume any strict priors that would limit the possible spectral response estimates and is thus applicable to any camera sensor, at least in the visible range. The estimates have low errors for estimating color channel values from known spectra, and are consistent with previously reported spectral response estimates.

## 1. Introduction

For using a camera as a measurement device, especially for measurements related to color, the knowledge of the camera system’s spectral response is essential. The spectral response, also known as spectral sensitivity, of the combined camera system, including lenses and filters, maps the spectral excitation to a measured per pixel value set. Individual pixels often represent the color channels red, green and blue. The spectral response can be used for various tasks, for example white balance adjustments [1,2], illumination estimation [3], and spectrum estimation in multi-spectral camera systems [4]. Some tasks, such as spectrum estimation in multi-spectral camera systems [5,6,7,8], are more sensitive to errors in the estimated spectral response versus the true spectral response. For such tasks, knowledge of the uncertainty of the estimated spectral response becomes important.

Color consistency estimation is an important task that would benefit from knowledge of the spectral response of the imaging system [9,10]. Humans have the innate ability to judge color in different lighting conditions [11], irrespective whether the object of color is outdoor in daylight or indoor under artificial white light. Computer vision methods lack this ability, and are hence possibly prone to errors in adversarial lighting conditions [12]. In addition, images of the same scene under the same lighting conditions but different equipment produce different color channel values. Color constancy correction based on known spectral response of the imaging setup would help computer vision models to overcome these problems [13].

This paper presents a new method for estimating the camera spectral response and its uncertainty. It is based on imaging a small number of color patches of known spectra and a small number of diffraction images produced by light dispersed through a diffraction grating, so that the complementary information provided by these two sources of information are integrated using a joint learning objective. The key novelty is in use of diffraction images for obtaining sufficiently high-dimensional data for spectral estimation without resorting to imaging monochromatic light at different wavelengths [14], whereas the color patches provide the required color consistency data as in standard methods [15,16]. By combining these two sources of information we can estimate the spectrum based on fewer color patches without needing to make strong assumptions on smoothness of the spectrum (that can be violated for example, for imaging systems that involve color filters). We use stochastic optimization [17] for estimating the spectral response parameters and quantify their uncertainty using a specific form of variational approximation [18,19,20]. The key advantages of the new method are the relative ease and speed of data acquisition, and the robustness of the spectral response estimates and the related uncertainties. Specifically, there is no need for expensive monochromator; the required equipment consists solely of low-cost and readily available components with device material costs not exceeding 100 euros, and a skilled individual can assemble it in a few hours.

## 2. Previous Methods

Previous methods for estimating the spectral response of a camera are typically based on a set of reflective color patches with known spectra [15,16] or using a monochromator for spectral response estimation [14]. Spectral estimation from a set of color patches relies on knowing the spectral distribution of the light source illuminating the patches and the spectral reflectivity of the imaged patches. The expression for the channel value $C_{c}$, where often c∈{R,G,B} that is, the red, green and blue channels, for wavelength λ, illuminant I(λ), patch reflectivity P(λ) and channel spectral response $R_{c}\left(\lambda \right)$ becomes
(1)$$\begin{array}{c} C_{c}=\int_{\lambda_{min}}^{\lambda_{max}}R_{c}\left(\lambda \right)P\left(\lambda \right)I\left(\lambda \right)d\lambda ,c\in \left\{R,G,B\right\}. \end{array}$$

Because of the digitization performed by the camera, Equation (Equation 1) can be written as a sum by discretizing each of the components in the integral and encoding dλ in $R_{c,\lambda }$, and assuming additive noise ϵ, as
(2)$$\begin{array}{c} C_{c}=\sum_{\lambda =\lambda_{min}}^{\lambda_{max}}R_{c,\lambda }P_{\lambda }I_{\lambda }+\epsilon ,c\in \left\{R,G,B\right\}. \end{array}$$

In principle, given enough measured patches the channel response $R_{c}$ can be solved by linear least squares fit for normally distributed noise ϵ. However, most natural spectra are low-dimensional [21], which causes the inversion problem to be ill-posed even with high number of patch measurements. Regularizing the inversion problem, for example, by using Tikhonov regularization [22] that encourages smooth channel response solutions, can yield good results for camera spectral response estimation [14], assuming that the underlying channel response is indeed smooth or low-dimensional [10]. Smooth solutions can also be encouraged by using a set of basis functions for spectral response estimation, for example Gaussian basis functions [23]. The smoothness assumption may be valid for many cameras, but for example when imaging through a color filter (external to the camera, not to be confused with the Bayer filter integrated onto the camera sensor) it can be violated. In such cases encouraging smoothness leads to poor results. In other words, we cannot always assume the spectra to be smooth.

A benefit of estimating the spectral response from color patches is the relative low cost of producing and imaging color patches, and the ability to capture calibration images outside laboratory conditions [24]. However, the reflected spectra depend on the illumination spectrum, and to achieve the best accuracy the spectrum of the illuminant needs to be known [24]. Differences between the assumed and true spectrum for the illuminant, for example for sunlight or a halogen light source, will cause errors in spectrum response estimates. It is also possible to achieve low-spectral-resolution spectrum response estimates using color patches without knowing the exact spectral power distribution of the illuminating light [25]. Assuming that no significant modifications to camera equipment are made and that for example no filters are used, spectral response estimation from color patches, as in, for example, Reference [10], may produce sufficiently accurate and high-resolution results.

The spectral response of a camera can also be estimated by measuring a camera’s response to monochromatic light [14]. Typically, the light is passed through a monochromator and its intensity is measured and associated with the camera’s response separately for each wavelength λ. The benefit of spectral response estimation using a monochromator is its algorithmic simplicity: the spectral response can simply be recovered by normalizing the camera’s response at a particular wavelength by the measured intensity of the light at that wavelength. To achieve high spectral accuracy, however, measurements need to be taken at many wavelengths. Again, a smoothness assumption of the spectral response can be made to reduce the required number of measurements. For example, one can measure the response at 10 nm intervals and estimate the remaining responses by interpolation.

The downsides of spectral response estimation using a monochromator are the high cost of monochromators, as well as the high manual effort required to measure a camera’s response at numerous wavelengths. If the number of measured wavelengths is reduced, the spectral accuracy of the spectral response suffers and may produce significantly inaccurate results to wavelengths between measurements points.

## 3. Method

Our approach builds on the common strategy of imaging color patches of known spectra, but complements this information with images of known illuminants taken through a diffraction grating, called diffraction images. To our knowledge, this is the first use of images of this nature for spectral response estimating.

Our estimation method takes as input 4 diffraction images and 11 color patch images. The purpose of the diffraction images is to provide sufficiently high-dimensional data for spectral response estimation in a way that is general enough for imaging setups that may include elements in addition to the camera and lens, for example color filters. The purpose of imaging color patches is to enforce consistency between measured color channel data and spectral response estimates. While it may be possible to capture a low-dimensional estimate of the spectral response using the color patches only [26], estimates based solely on color patches appear to be inaccurate; our attempts to recover spectral response estimates using color patches only yielded noisy estimates without heavy smoothing or regularization. The diffraction images would in principle be sufficient for estimating the full spectral response even without the color patches, but in practice the estimates produced solely based on the diffraction images are not sufficiently robust for high-dimensional spectral responses. By combining these two types of information sources, however, we can reliably estimate the response already based on the small number of calibration images.

In the following, we first describe the imaging setup for diffraction images in Section 3.1 and then explain the details of the two input data sources and their preprocessing in Section 3.2–Section 3.4. Finally, in Section 3.5 and Section 3.6 we explain the algorithm for estimating the spectral response and its uncertainty.

### 3.1. Imaging Setup for Diffraction Images

The imaging setup for capturing diffraction images is show in Figure 1, and an example image produced by the setup is show in Figure 2. The imaging setup consists of an illuminant, a diffuser, an optional color filter, a narrow slit, a diffraction grating and the camera system. The part between the slit and the camera is enclosed and the interior is painted black to minimize reflections and external light contamination. The illuminant provides a source of light for the imaging setup. The purpose of the diffuser is to provide even illumination entering the slit, and an optional color filter can be used to filter the light of the illuminant to shape the spectrum. The diffraction grating used is a linear diffraction grating made of PET plastic film, with a marketed diffraction grating constant of 500 grooves per millimeter. The angle between the incident ray of light from the slit and the first order diffraction component is determined by the grating equation $\sin(\theta_{1})=n\lambda$ where *n* is the grating constant, and thus the angle between the two enclosures before and after the diffraction grating should be designed according to the grating constant.

### 3.2. Diffraction Images

The general idea of capturing diffraction images is to collect data of an illuminant, with a known relative spectral distribution $v^{S}$, dispersed into the illuminant’s spectral components as seen by the imaging system that is the subject of spectral response estimation. This is accomplished by passing the light of an illuminant through a narrow slit and the resulting ray of light is passed through a diffraction grating. The diffraction grating disperses the light into its spectral components and can then be imaged using the camera system for which we wish to find the spectral response for. All diffraction images were taken from the same location with respect to the imaging setup so that wavelength locations remained constant for each image. The resulting diffraction image resembles an image of a rainbow as seen in Figure 2.

We repeat the procedure of capturing diffraction images for three different illuminant spectra denoted by $v_{j}^{S}$, where *j* is the illuminant index. The spectra of each of the three illuminants are either measured or known, for example for a known black body illuminant, so that the known spectra can be used for later processing. We use three illuminants, because we wish not only to find the spectral response of the camera system, but to separate the spectral attenuation caused by the system external to the imaging system, that is, the diffraction grating efficiency. In principle, two illuminating spectra should be enough to solve for two unknowns for each wavelength (spectral response and diffraction grating efficiency), but due to anticipated noise in the data and difficulties in finding suitable illuminants that extend over the assumed spectral range of 380 nm to 720 nm (especially the wavelength range between 380 nm to 420 nm, for which the relative intensity of many white LED sources is very low or may include significant troughs), three illuminating spectra are used.

The first illuminant is a halogen light source. The second and third illuminants use the same light source filtered through purple and blue light filters commonly used to filter the light of camera flashes. These filters are chosen because they attenuate light selectively in the range of 380 nm to 720 nm, but do not entirely block any of the wavelengths. Furthermore, the three illuminating spectra are chosen, because they are anticipated to produce enough diversity in the measured data. The relative spectral distribution of the three illuminants can be seen in Figure 3.

A fourth image is captured using the same imaging setup, but with two laser light sources: one at the blue end of the spectrum and another at the red end of the spectrum, as seen in Figure 4. For the purpose of the method, these laser sources produce sufficiently monochromatic light so that a wavelength to pixel location mapping can be computed. This wavelength to pixel location mapping allows us to extract the relative intensity of light incident on the camera sensor for a narrow wavelength band. The central wavelength of the lasers is measured and in our experiments were 409 nm (blue/purple) and 638 nm (red). The center wavelengths were measured for each camera measurement individually, but the central wavelength remained stable at the stated wavelengths. For each row in the produced image the location of the maximum intensity for the blue and red are mapped to the measured laser central wavelength. All other wavelengths are interpolated by assuming a linear mapping from wavelength to pixel location. The angle between the wavelengths 380 nm and 720 nm from the diffraction grating is assumed to be small, so the small angle approximation of the diffraction equation is justified: sin(θ)=nλ is approximated by θ=nλ, where θ is the diffraction angle from the normal of the diffraction grating of the *n*-th diffraction component, and λ is the wavelength of light. For the following, the diffraction order *n* is assumed to be 1.

### 3.3. Color Patch Images

The color patch images are captured using a set of transmissive color filters, through which an adjustable light source is illuminated. This capture method is in contrast to the more common approach of imaging reflective patches, but results in the same effect with the benefit that patch spectra can be measured using a spectral illuminance meter. The adjustable light source used is a set of 11 LEDs with different spectra. 16 different transmissive color filters are used, yielding a total of 176 different color patches and spectra. An example of a patch image is shown in Figure 5, and normalized patch spectra are presented in Figure 6. The spectra of each patch are individually measured to avoid possible error caused by fluorescence (as opposed to calculating the spectra produced by the combination of a LED illuminant and the transmission properties of each color filter), using a Hopoocolor OHSP-350C spectral illuminance meter with a reported wavelength accuracy of ±0.5 nm and FWHM of 2 nm. The images of the patches produce varying size pixel areas, for which the mean value of each patch and each channel is computed for later processing. For a given channel spectral response and for a given patch spectrum, the channel values can be computed using Equation (Equation 2), where the spectrum $P_{\lambda }^{S}$ of produced by a patch replaces the product term $P_{\lambda }I_{\lambda }$.

### 3.4. Data

After averaging over the rows, the data for the diffraction images are in the form of vectors—one for each image: $v_{i,j,c}^{M}\in R_{\geq 0}^{n_{\lambda }},\forall i,j,c$, where $n_{\lambda }$ is the number of wavelength sampling points in the vector, *i* is the camera index and *j* is the illuminant index (the *M* superscript is used to denote monochromatic data collected in the diffraction images). For the assumed spectral range of 380 nm to 720 nm, we have $n_{\lambda }=340$. Similarly, for each $v_{i,j}^{M}$ we have a single measured spectrum for the illuminant $v_{j}^{S}\in R_{\geq 0}^{n_{\lambda }},\forall j$ where the measured illuminant spectrum is independent of the channel, that is, same for red, green and blue channels. The measured spectra, $v_{j}^{S}$ are all individually normalized by dividing by the value for λ=550 nm, which is chosen because it is halfway between the assumed limits of spectral responses. Similarly, the measured vectors $v_{i,j}^{M}$ from the diffraction images are normalized by dividing by $v_{i,j,\lambda =550\;\mathrm{nm}}^{M}$ for all *i* and *j*, which corresponds to the green channel’s value at 550 nm for each camera and each illuminant.

Each patch image is normalized by dividing all raw image values by the exposure time of that image, because the aperture and ISO settings are kept constant for the patch photos of one camera. The patch regions are extracted from the images and the median value for each channel of each patch is produced. These median values for each channel of each patch are then used for parameter estimation—$p_{i,J,c}$ where *i* is the camera index, *J* is the patch index and c∈{R,G,B}. Finally, all patch values for a camera are normalized by diving by the maximum channel value of all patches, such that all resulting channel patch values are between 0 and 1.

### 3.5. Parameter Estimation

The objective is to estimate the spectral responses of a camera or a set of cameras: spectral response $R_{i,c}\in R_{\geq 0}^{n_{\lambda }},\forall i,c$ for camera *i* and channel *c*. We are interested in finding the relative spectral responses for all channels, relative to the green channel’s spectral sensitivity at 550 nm. The imaging setup, as depicted in Figure 1, which produces monochromatic rainbow-like images seen in Figure 2, has a linear diffraction grating with an unknown grating efficiency, that is, the ratio of input power to the ratio of output power as a function of wavelength $\eta \left(\lambda \right)=\frac{P_{out}\left(\lambda \right)}{P_{in}\left(\lambda \right)}$. The grating efficiency is an unknown property of the calibration setup that is constant for all illuminants and camera, but it needs to be solved, or known or assumed a priori, to yield correct results for the spectral responses. The discretized grating efficiency is denoted by $\eta_{\lambda }$ for wavelength λ. The grating efficiency is modeled by a quadratic polynomial $\eta \left(\lambda \right)=\sum_{q=0}^{2}a_{q}\lambda$, which enforces the grating efficiency to be smooth and helps mitigate possible overfitting. Overfitting was observed to be a problem if each $\eta_{\lambda }$ are modeled freely.

The parameters $R_{i,c,\lambda }$ (for all triplets of i,c,λ) and $a_{q}$ (for all *q*) are estimated by stochastic gradient descent using ADAM optimizer [17] against an overall loss:$$\begin{array}{cc} L_{total} & =\alpha L_{mon}+\beta L_{patch}+\gamma \sum_{\lambda =\lambda_{min}}^{\lambda_{max}}\Gamma_{\lambda }, \end{array}$$
which is a sum of monochrome data loss $L_{mon}$, color patch loss $L_{patch}$, and a regularizing term for the grating efficiency $\Gamma_{\lambda }$:$$\begin{array}{cc} L_{mon} & =\frac{1}{3n_{\lambda }n_{cameras}n_{illuminants}}\sum_{\begin{array}{c} i,j,c,\lambda \end{array}}{\left(R_{mon}^{i,j,c,\lambda }\right)}^{2}\\ R_{mon}^{i,j,c,\lambda } & =v_{i,j,\lambda }^{M}-R_{i,c,\lambda }\eta_{\lambda }v_{j,\lambda }^{S}\\ L_{patch} & =\frac{1}{3n_{cameras}n_{patches}}\sum_{\begin{array}{c} i,J,c \end{array}}{\left(R_{patch}^{i,J,c}\right)}^{2}\\ R_{patch}^{i,J,c} & =p_{i,J,c}-\sum_{\lambda =\lambda_{min}}^{\lambda_{max}}R_{i,c,\lambda }P_{J,\lambda }\\ \Gamma_{\lambda } & =\left\{\begin{array}{cc} |\eta_{\lambda }-t\;\cdot \;max\left(\eta \right)|, & \mathrm{if}\;\eta_{\lambda }\leq t\;\cdot \;max\left(\eta \right)\\ 0, & \mathrm{otherwise}. \end{array}\right. \end{array}$$

The residual terms for the monochromatic data $R_{mon}$ and color patch data $R_{patch}$ enforce the solution to be consistent with the measurement results. The regularization term for diffraction grating efficiency $\Gamma_{\lambda }$ enforces that the diffraction grating efficiency should be at least t·max(η) for all wavelengths, where *t* is chosen to be 0.2. This is based on visual observations that there does not appear to be any wavelength in the visible range at which the intensity falls significantly yet allowing for great enough range. The weighing terms α and β are both set to unity for $L_{total}$ for equal weighting of monochrome and patch data. The parameter γ controls the amount of regularization by $\Gamma_{\lambda }$ and is set to 0.1. The hyperparameters α, β and γ could be tuned for a better fit or to better match the subjective importance of each of the terms, but in our experiments the default choices mentioned above were found to be satisfactory.

All spectrum response values, $R_{i,c,\lambda }$, are initialized by draws from the random uniform distribution between 0 and 0.1. The intercept value of the polynomial for the grating efficiency is initialized to 1, and other polynomial coefficients are initialized by draws from the random uniform distribution between −1 and 0. After each gradient-training epoch, all negative values of $R_{i,c}$ and $\eta_{\lambda }$ are set to zero to enforce positivity.

### 3.6. Uncertainty Estimation

Besides learning the best possible camera response parameters $\theta =\{R_{i,c},\eta_{\lambda }|\forall i,c,\lambda \}$, we are interested in quantifying the uncertainty of this estimate, in order to recognize whether the response have been estimated sufficiently accurately, as well as to estimate the uncertainty of the predicted RGB channel values. This can be formally done by inspecting the posterior distribution p(θ|D), where $D=\{v_{i,j,\lambda }^{M},p_{i,J,c}|\forall i,j,J,c\}$ is all of the data available for estimating the parameters. According to the Bayes’ rule we have p(θ|D)∝p(D|θ)p(θ), but for our case we do not have obvious physical grounds for determining the likelihood p(D|θ) or the prior p(θ). Nevertheless, we can obtain useful uncertainty estimates by local approximation for the posterior at the mode (the parameter estimate explained in Section 3.5). For the implementation we use software provided by Reference [27].

Conditional on the mode estimate, denoted by $\mu_{\theta }$, we estimate the uncertainty following the variational inference procedure of Reference [18]. We use a Gaussian prior $\theta ∼N(\mu_{\theta },\Sigma_{\theta })$ where $\Sigma_{\theta }$ is a diagonal matrix, corresponding to independence of the parameter dimensions. After learning $\mu_{\theta }$ we complement the loss with a Kullback-Leibler divergence $L_{KL}$
$$\begin{array}{cc} L_{KL} & =\frac{1}{n_{parameters}}\sum_{\begin{array}{c} l \end{array}}\left(\frac{\sigma_{l}^{2}+\mu_{l}^{2}}{2\sigma_{p}^{2}}-log\left(\frac{\sigma_{l}}{\sigma_{p}}\right)-\frac{1}{2}\right)\\ \sigma_{p} & =\left\{\begin{array}{cc} \frac{1}{\sqrt{3\left(\lambda_{max}-\lambda_{min}\right)n_{cameras}n_{illuminants}}}, & \mathrm{for}\;R_{i,c}\\ \frac{1}{\sqrt{\lambda_{max}-\lambda_{min}}}, & \mathrm{for}\;\eta \end{array}\right., \end{array}$$
where $\sigma_{l}$ are the posterior variances, $\mu_{l}$ are the estimated parameters, and $\sigma_{p}$ are the prior variances (diagonal elements of $\Sigma_{\theta }$). Summation for $L_{KL}$ is over all the parameters.

To estimate the variance we follow the procedure of References [19,20] for quantifying predictive uncertainty of neural networks; our model can be interpreted as very shallow network that is trained with stochastic gradient descent. We train an ensemble of *N* response estimates with different random initial conditions, estimating the mean parameters $\theta_{\mu }^{n}$ for each iterate. We then form the estimate $\hat{\mu_{\theta }}=\frac{1}{N}\sum_{n}\mu_{\theta }^{n}$ that is used as the mean parameter for estimating the variances $\sigma_{\theta }^{n}$, separately for each iterate, and the final posterior estimate for θ becomes $\hat{\theta }∼N\left(\hat{\mu_{\theta }},\hat{\sigma_{\theta }}\right)$, where $\hat{\mu_{\theta }}=\frac{1}{N}\sum_{n}\mu_{\theta }^{n}$.

## 4. Experimental Results and Methods

To test our method, we measured the spectral response of four cameras: Canon 300D, Canon 10D, Nikon D300S and Nikon D300. We chose these cameras because earlier ground truth measurements for these cameras were made by References [1,10], and spectral response data (for visual comparison) can be found in References [28,29] respectively. Data from these sources will be used as ground truth for spectral response comparisons and spectral response error quantification.

### 4.1. Experimental Methods

For comparison to the proposed method, we repeat the spectral response estimation against a baseline method and four variations of the proposed method. The proposed method uses all the data, monochromatic and patch, and includes diffraction grating efficiency estimation. As the baseline method, we calculate the spectrum response to be the mean of illumination spectrum corrected, that is, divided by the illuminant spectral distribution, monochromatic data. The baseline assumes constant diffraction grating efficiency for all wavelengths. We also compared to a reduced version of our method, called *monochromatic only*, where we do not consider the color patch data, that is, β=0 in $L_{total}$. Both *constant grating* and *monochromatic only, constant grating* are methods in which the diffraction grating efficiency is assumed to be one for all wavelengths. Furthermore, in *monochromatic only, constant grating*, the color patch data is not considered, that is, β=0. Finally, we compare against variant of the method that uses the color patch data only, called *color patches only*, which uses an augmented loss function to regularize the solution.

To quantify the accuracy of the resulting spectral responses for each of the methods being compared, we compare against measured RGB channel values and against ground truth spectral responses. For RGB errors we calculate the residuals of the normalized RGB channel values, where the normalization is done same as described in our method, that is, divided by the maximum channel value for each camera such that all channel values are between 0 and 1 and remain proportional to each other. The mean of spectral responses is used to calculate RGB channel value predictions in the baseline, whereas the mean of ensemble results is used in the remaining methods. Because of the scaling of measured and estimated channel values, the mean absolute errors for all the cameras diminish considerably if lower channel values are not considered in error estimates. This indicates that the relative error for small values is greater than for larger values, where larger values correspond to brighter objects or increasing exposure.

### 4.2. Results

The main results of the comparison are reported in Table 1 and Table 2. Table 1 shows RGB errors with method variant *color patches only* producing the lowest RGB errors. The hyperparameters of the method variant *color patches only*, described in Section 4.2.4, are optimized against the ground truth, because the method variant requires additional regularization of the spectral response estimate not required by the other method variants. Of the methods that do not require additional regularization of the spectral response estimate, the proposed method outperforms the alternatives for three of the four cameras, with average error reduction of 26% (0.16 vs. 0.22) compared to the baseline and a slight improvement (0.16 vs. 0.17) over the ground truth. Table 2 shows the mean spectral response error of the different methods, when compared to ground truth data, and demonstrates that the proposed method outperforms, on average, the other compared methods. The results for ground truth in Table 1 have been calculated using the patch spectra and ground truth spectral response data for the available range of 400–700 nm. Predicting RGB values using ground truth data also performs well, with the largest error for Canon 10D. The mean spectral response error in Table 2 has been calculated between the range 400–700 nm as ground truth data is only available in this range. Errors in Table 2 have been calculated between the mean of ensemble iterates and ground truth data. In the following subsections we analyze the results of these three methods in more detail and provide visual illustrations. Further analysis of *constant grating* and *monochromatic only, constant grating* is omitted as they produce, on average, worse results that the proposed method and *monochromatic only*.

#### 4.2.1. Baseline

The results of the baseline are shown in Figure 7e–h, overlaid with ground truth spectral responses obtained from Reference [28] for Canon 300D and Nikon D300S, and from Reference [29] for Canon 10D. The baseline assumes the diffraction grating efficiency of diffraction grating used in the diffraction image formation apparatus to be constant for all wavelengths. A visual comparison of the baseline results against ground truth reveals the functional forms of the individual channel spectral responses to be very similar. However, there is a significant difference between the relative maximum intensities of different channels with a largest difference between the blue and green channels. Some of the differences between the maximum relative sensitivities, as compared to the ground truth, maybe be explained by the difference sampling resolutions. Although the Nikon D300S and Nikon D300 are assumed to be built around the same sensor model, there are significant differences between the relative channel sensitivities. There is also a pronounced difference between the two cameras in the red channel, for which the Nikon D300 shows significant sensitivity also in the range below 550 nm, which is consistent with the ground truth, whereas the response result in Figure 7g does not show similar response to wavelengths below 550 nm. While the two cameras may have the same sensor models, the subsequent analog-to-digital conversion or the infrared filters may have differences. There may indeed even be differences in spectral sensitivities between different units of the same camera model. Note, however, that for image capture for both the Nikon D300S and Nikon D300 the exact same optics were used.

#### 4.2.2. Monochromatic Only, β=0

The spectral response estimates for *monochromatic only* are shown in Figure 7i–l and the diffraction grating efficiency estimate is show in Figure 8b. For each camera, all ensemble iterates are plotted, which show moderate amount of uncertainty for each camera and for each color channel. For all the cameras, the green channel’s spectral response estimates demonstrate the lowest amounts of variance, which can be partially explained by normalization by the green channel’s value at 550 nm and because the green channel has the largest total spectral response. The peak relative spectral responses for different channels appear to be closer to the ground truth than in the baseline, with the exception of Canon 300D, for which also the estimated error for *monochromatic only* is larger than for the baseline as presented in Table 1. Overall, using only monochromatic data along with presented method improves on results compared to the baseline method and also provides a better estimate for the error in the spectral responses for each channel and wavelength. While the *monochromatic only* method variant performs slightly worse that the proposed method, it only needs 4 diffraction photos and not the 11 patch photos and therefore requires less work and equipment.

#### 4.2.3. Proposed, All Data

The spectral response estimates for the proposed method are shown in Figure 7a–d and the diffraction grating efficiency estimate is show in Figure 8a. Similar to *monochromatic only*, 2500 ensemble iterates are trained and plotted. The estimates produced by the proposed method are visually very similar to *monochromatic only*, but generally seem to recover areas of channel spectral responses where the spectral sensitivity is low better than in *monochromatic only*. The error results as estimated based on the color patch residuals are better than for the baseline and for *monochromatic only*, no less because the proposed method is also trained against the square of color patch residuals. The purpose of including the color patch data in the training is to drive the estimated spectral responses to be consistent with measured RGB-channel values.

The results in Table 2 show that the proposed method produces better results for most cameras, and on average, than the other methods being compared. Results in Table 1 also confirm that the proposed method does benefit from using the combination of color patches and diffraction images. For the results in Table 2, it is notable that the ground truth results are of significantly lower resolution, so even a perfect fit would produce comparative error due to differences in spectral resolution. For example, Figure 7c,d show a very close match to the ground truth, yet the oscillation in the results produced by the proposed method causes comparative error. The oscillation seen in the results is also present in the raw images, which is observable by close examination of Figure 2, which has been captured using a Nikon D300 camera.

#### 4.2.4. Color Patches Only

Spectral response estimates using color patch data only are show in Figure 7m–p. These estimates have been obtained by optimizing the spectral response parameters $R_{i,c,\lambda }\forall i,c,\lambda$ against the augmented loss $L_{total,CPO}=\beta L_{patch}+L_{CPO}$, where $\beta L_{patch}$ is as before, and the other terms are
$$\begin{array}{cc} L_{CPO} & =\omega_{1}S_{1st}+\omega_{2}S_{2nd}+\omega_{3}S_{sparse}+\omega_{4}S_{bound}\\ S_{1st} & =\sum_{i,c,\lambda =\lambda_{min}}^{\lambda_{max}-1}\left|R_{i,c,\lambda +1}-R_{i,c,\lambda }\right|\\ S_{2nd} & =\sum_{i,c,\lambda =\lambda_{min}+1}^{\lambda_{max}-1}\left|R_{i,c,\lambda -1}-2R_{i,c,\lambda }+R_{i,c,\lambda +1}\right|\\ S_{sparse} & =\sum_{i,c,\lambda }R_{i,c,\lambda }\\ S_{bound} & =\sum_{i,c}\left(R_{i,c,\lambda =\lambda_{min}}+R_{i,c,\lambda =\lambda_{max}}\right), \end{array}$$
with hyperparameters $\omega_{1}=2.325\times 10^{-5}$, $\omega_{2}=1.473\times 10^{-4}$, $\omega_{3}=3.208\times 10^{-9}$, and $\omega_{4}=1.034$.

Spectral response estimation based on only the color patches is severely ill-posed, and augmenting the total loss with $L_{CPO}$ serves as a regularizer to yield informative solutions. Similar to the regularization in Reference [24], $S_{1st}$ and $S_{2nd}$ enforce solution smoothness, $S_{sparse}$ enforces sparsity, and $S_{bound}$ enforces that the spectral response solutions be close to zero at $\lambda_{min}$ and $\lambda_{max}$. The values for the hyperparameters were found using Bayesian optimization [30], optimized to minimize the mean of mean absolute errors between estimated and ground truth spectral responses.

It is important to note that this baseline requires access to ground truth spectral responses for some cameras in order to find suitable regularization parameters, which would be problematic in practical use, and in our case it was allowed to use the ground truth responses for the same cameras (but only for determining the hyperparameters). Despite requiring the additional data, the method is not able to replicate the spectral responses well (Table 2), which clearly demonstrates the value of the diffraction images. In terms of RGB errors (Table 1), however, it achieves the smallest error. Without regularization it would reach even smaller RGB error (mean of 0.08569 vs. 0.1285 with regularization), but the corresponding spectral responses were extremely poor, showing several spurious peaks most likely due to overfitting to patch spectra.

#### 4.2.5. Uncertainty

The results for the relative uncertainty in spectral responses, σ related to $R_{i,c}$, in relation to the wavelength λ are the same for all cameras and color channels. Furthermore, the relative uncertainty is the same for each of the different method variants, and appears only to be a function of the wavelength λ. This leads to the conclusion that uncertainty is inherent to the method. Only small differences in the relative uncertainty with respect to λ are observed, such that uncertainty is greatest at small λ, where it is at most approximately 1.6% of the value at 550 nm, and smallest at longer wavelengths. This suggests that the uncertainty is mainly caused by the relative differences in the spectral distribution of the illuminants used to generate diffraction images. The uncertainty is greatest at wavelengths where the intensity of the light is lowest, and vice versa. An ideal condition would be for the uncertainty in the spectral responses to be generally low and constant for all wavelengths. A way to achieve this might be to use an illuminant with a flat spectral distribution over the measurement wavelengths. Another way could be to capture images with different exposure settings, for example exposure time, for different parts of the spectrum.

## 5. Conclusions

The practical camera spectral response and uncertainty estimation method presented here produces high-resolution spectral response estimates with only a few images. The method does not rely on any smoothness or dimensionality priors restricting the spectral sensitivity estimates and is thus generally applicable to a camera combined with any color filter that selectively attenuates spectra. The method relies on first obtaining high dimensional estimates of the channel spectral responses and imposing consistency with imaged channel values, and simultaneously solving for the diffraction efficiency of the used diffraction grating. The obtained mean channel spectral response estimates are usable for estimating relative channel values from given spectral excitation. Furthermore, the distribution of estimates can be used to estimate the error distribution associated with estimated relative channel values improving robustness of computer vision models for example against adversarial lighting conditions [12]. The produced spectral response estimates between cameras are in relation to each other and therefore the estimates can be used for color consistency corrections for models trained on input from one camera and inferred using input from another [13]. Finally, the uncertainty estimates of spectral sensitivities indicate a strong relationship between the spectra of the illuminants used as calibration targets and provide an understanding on how to reduce relative uncertainty in calibration results.

## Figures and Tables

**Figure 1 jimaging-06-00079-f001:**
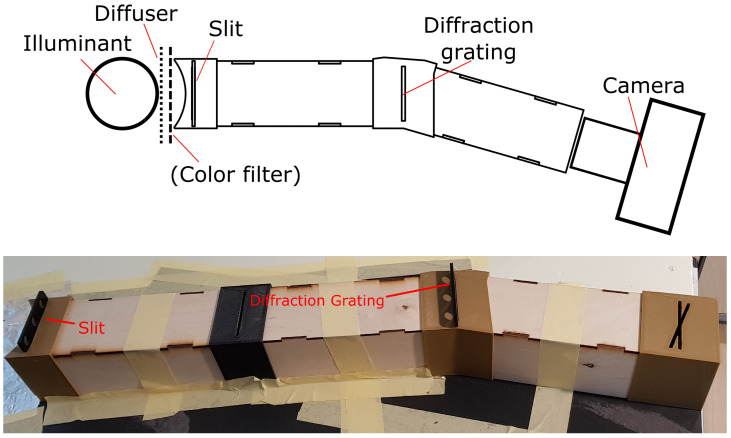
Overhead schematic (top; not to scale) and photo (bottom) of the imaging setup consisting (from left to right) of the illuminant, diffuser, an optional color filter, narrow slit, an enclosure, diffraction grating, an enclosure and the camera system being measured. The photo shows only the enclosed equipment, omitting the illuminant, the diffuser and the camera.

**Figure 2 jimaging-06-00079-f002:**
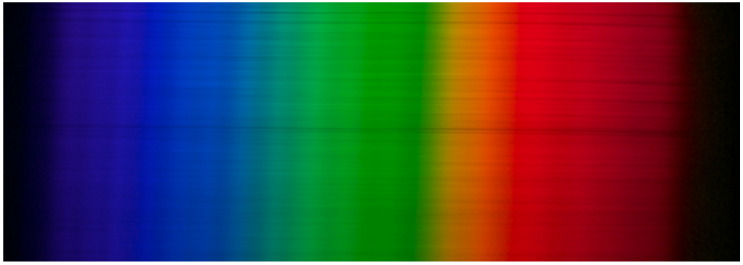
An example diffraction image of a range of monochromatic light, imaged by Nikon D300 camera.

**Figure 3 jimaging-06-00079-f003:**
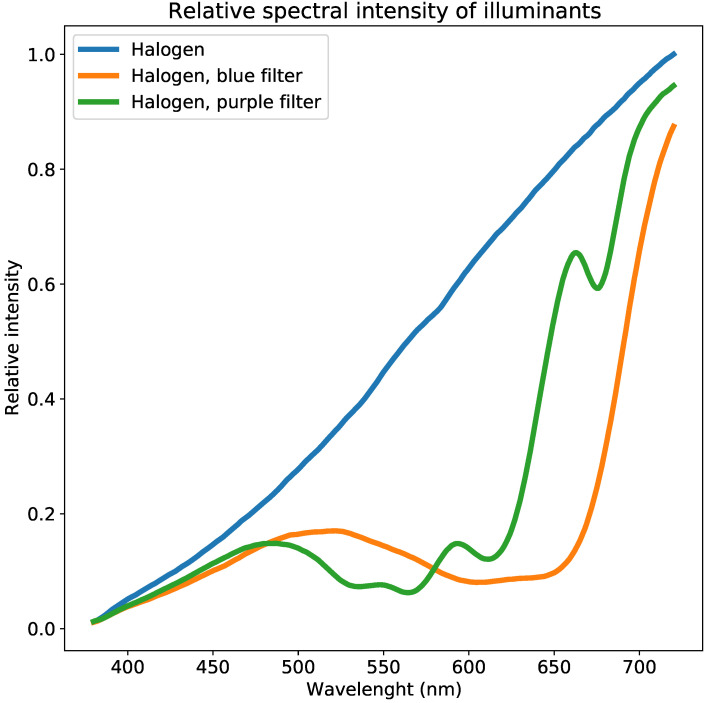
Spectra of the illuminants used to generate diffraction images as presented in Figure 2. The equipment is unable to estimate sensor sensitivity below wavelengths of approximately 380 nm, due to all illuminants having very low intensity.

**Figure 4 jimaging-06-00079-f004:**
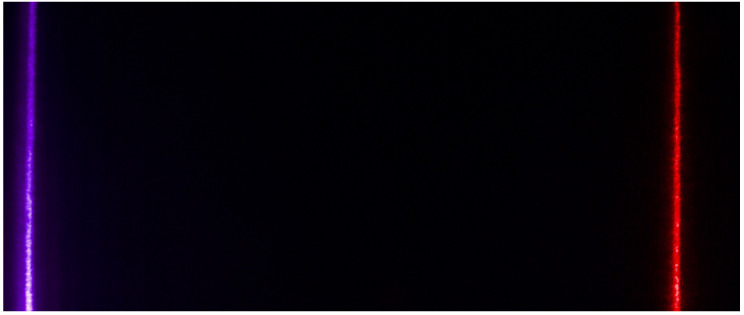
An example image of two laser sources imaged through the imaging setup, showing lines at 409 nm (blue/purple) and 638 nm (red).

**Figure 5 jimaging-06-00079-f005:**
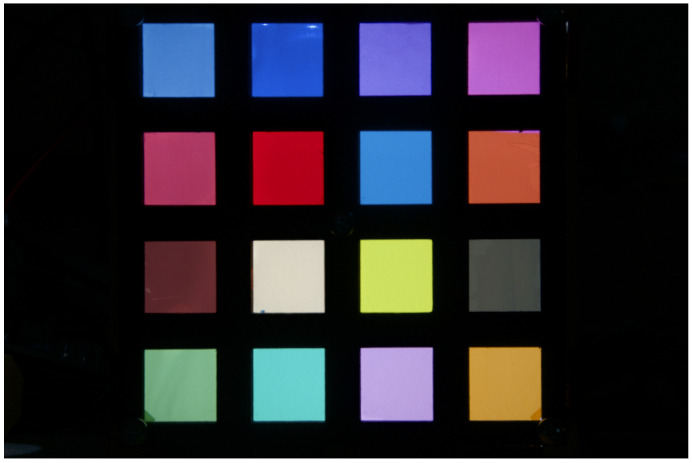
An example image of a color patch photo. The color patches are transmissive color filters used for filtering the light of camera flashes, illuminated from behind using a LED producing white light at correlated color temperature of 10,833 K.

**Figure 6 jimaging-06-00079-f006:**
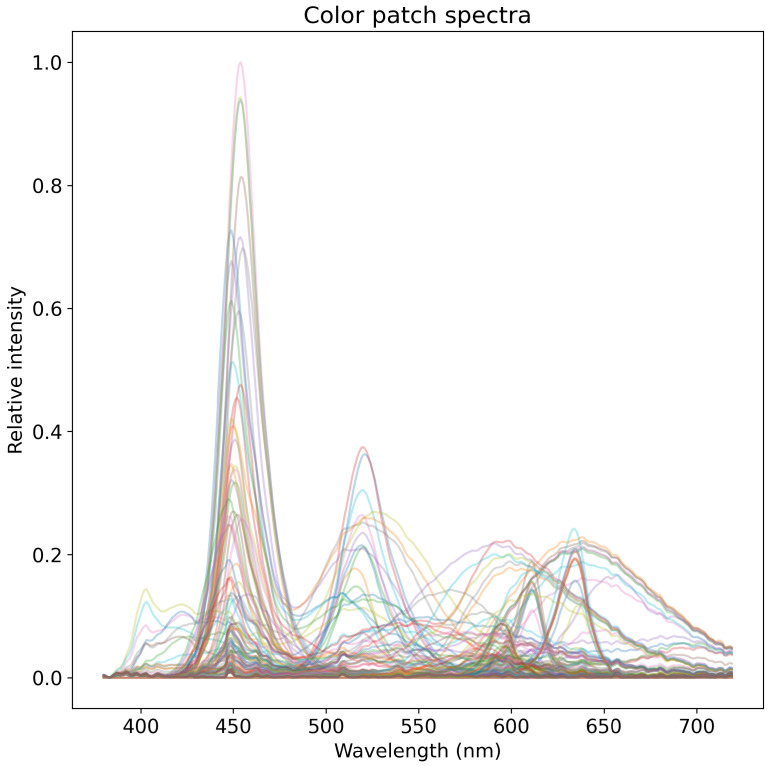
Plot of all spectra of color patches normalized by the maximum intensity. The blue of white LEDs dominates near 450 nm, being more pronounced for cold white LEDs with a relatively higher correlated color temperature. Spectra are colored according to illuminating LED source.

**Figure 7 jimaging-06-00079-f007:**
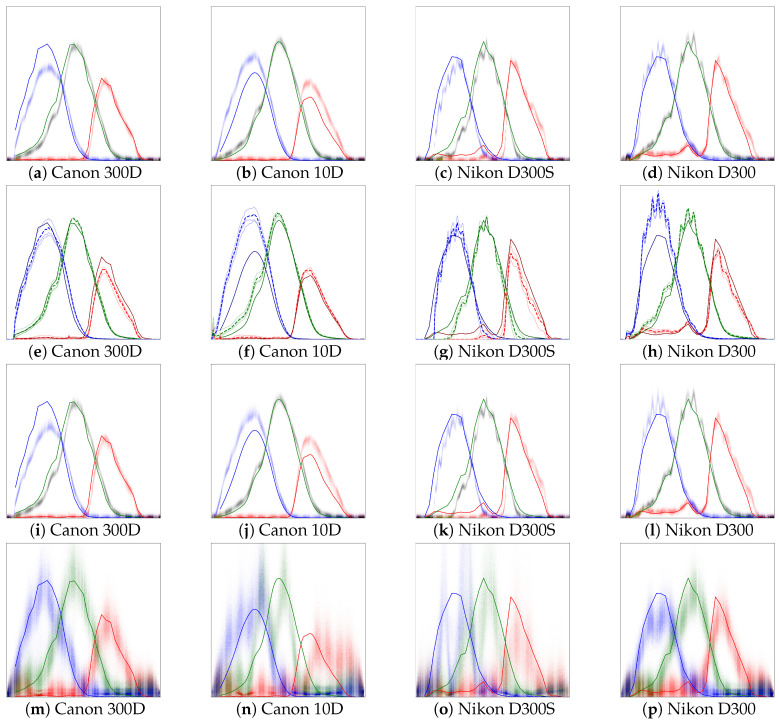
Spectral responses for proposed (**a**–**d**), baseline (**e**–**h**), monochromatic only (**i**–**l**), and color patches only (**m**–**p**), all over the range of 380–720 nm. The solid line corresponds to the ground truth measures using a monochromator, obtained from References [28,29]. Spectral response results have been scaled by diving all values by the value for the green channel at 550 nm. All ensemble iterates for proposed, monochromatic only, and color patches only methods have been plotted. For the baseline, results for all illuminants have been plotted and also the mean estimate from the three illuminants as dashed line. For Nikon D300S and D300 the results may seem noisy, but the oscillation visible in the spectral responses is also visible in the raw diffraction images. For clarity, the opacity of ensemble iterate points in (**m**–**p**) is five times that of the other sub-figures.

**Figure 8 jimaging-06-00079-f008:**
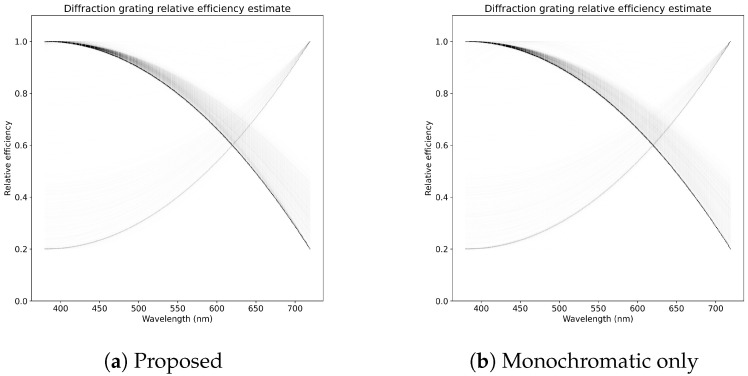
Plot of all ensemble iterates of the diffraction grating efficiency estimate for proposed and *monochromatic only* methods. There are slight differences in the distribution of the samples over the wavelengths, but, overall, these methods produce estimates that are consistent with each other. A small number of ensemble iterates have converged to an optimum where the grating efficiency is greater for longer than smaller wavelengths.

**Table 1 jimaging-06-00079-t001:** Mean of absolute RGB channel relative residuals, relative to the mean of measured channel values for each camera, are presented here. Of all the considered variations of the method, the proposed method produces the second lowest channel errors in estimating the relative channel response—approximately 27% lower errors compared to the baseline. The lowest errors are produced using the *color patches only* method, which, however, requires ground truth spectral responses for training. Bolded values indicate lowest error for each camera.

RGB Errors					
Method	Canon 300D	Canon 10D	Nikon D300S	Nikon D300	Mean
**Proposed**	0.1254	0.1965	0.1766	0.1347	0.1583
Baseline	0.1234	0.2131	0.2832	0.2486	0.2171
Monochromatic only	0.1529	0.1970	0.2036	0.1629	0.1791
Constant grating	0.1219	0.2109	0.2166	0.2091	0.1896
Monochromatic only, constant grating	0.1252	0.2137	0.2864	0.2468	0.2180
Color patches only	**0.1137**	**0.1820**	**0.1349**	**0.0832**	**0.1285**
Ground truth	0.1192	0.2360	0.1629	0.1488	0.1667

**Table 2 jimaging-06-00079-t002:** Mean absolute error of spectral response compared to ground truth. Errors have been calculated between the range 400–700 nm as ground truth data is only available for this range. Of all the compared methods, the proposed method produces the lowest mean error. Bolded values indicate lowest error for each camera.

Spectral Response Errors					
Method	Canon 300D	Canon 10D	Nikon D300S	Nikon D300	Mean
**Proposed**	0.0522	**0.0506**	0.0580	**0.0346**	**0.0489**
Baseline	**0.0385**	0.0672	0.0625	0.0574	0.0564
Monochromatic only	0.0546	0.0510	0.0603	0.0366	0.0506
Constant grating	0.0453	0.0586	**0.0556**	0.0505	0.0525
Monochromatic only, constant grating	0.0464	0.0587	0.0598	0.0574	0.0555
Color patches only	0.0586	0.1298	0.1148	0.0638	0.0918

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
