# Peer review of "Practical Camera Sensor Spectral Response and Uncertainty Estimation"

_2313-433X, 2020, doi:10.3390/jimaging6080079_

Round 1

Reviewer 1 Report

This paper presents a method for estimating camera spectral sensitivity using diffraction images and color patch images. The key of the method is to use diffraction images, which are captured using a diffraction grating, to obtain high-spectral-resolution data to complement standard three-channel color patch data.

The use of the diffraction grating is novel for camera sensitivity estimation and it enables a more low-cost setup compared with the system based on monochromatic light. However, the accuracy improvement by using the diffraction grating, compared with a standard color patch-based method based on a linear basis model (e.g., [10]), is not experimentally clarified in the paper.

The followings are detailed comments that should be addressed in the revised version.

1) I could not fully understand what the numerical values evaluated in Table I are. Are they RGB value errors for color patches? Please clarify. If I assume that Table I evaluates the RGB errors, the numerical evaluation for camera spectral response estimation errors (in the wavelength domain) should be added to directly evaluate the accuracy improvement of the estimated camera spectral functions by using the proposed method.

2) Since all the compared methods use diffraction images, it still remains unclear how much the estimation accuracy is improved by exploiting the diffraction grating, in comparison with a standard method using only color patches. To confirm the effectiveness of using the diffraction grating, the comparison with the baseline method using color patch data only (with known illumination spectra and a linear basis model for the camera responses, e.g., [10]) should be added. (This method may correspond to "color patch only" in the author's method.)

3) Also, it is hard to judge and compare the estimation accuracy from Figure 7. Each estimation result should be overlaid with the ground truth.

4) Throughout the paper, the authors use the term “color filter” to represent an optional color filter placed in front of the camera system. However, it is a little confusing because “color filter” is also commonly used to represent on-sensor RGB filters to form RGB responses (i.e., R_c). Thus, a more careful explanation is necessary for the usage of “color filter” in the paper.

5) In addition to the above comment, the explanation “the color patch images are captured using a set of color filters” in line 159 is also a little confusing because the color patch is usually considered as a reflective patch (e.g., X-Rite ColorChecker). Thus, a more careful explanation is recommended. Also, Figure 5 should be cited in this subsection.

6) The explanation of the optimization cost and procedure in section 3.5 should be improved.

 - In line 198-199, there is the description that the grating efficiency is modeled by a quadratic polynomial. However, there is no mathematical explanation for the specific form of the used model. Thus, optimization parameters for the grating efficiency remain unclear.

- For a series of equations on page 8, it could be clearer that the authors first show the final optimization cost “L_total” and then explain each term by introducing the detailed meaning of mathematical symbols.

- Please clarify the optimization parameters to be estimated to minimize “L_total”.

7) Two related papers are recommended to include.

 - Quick Approximation of Camera’s Spectral Response from Casual Lighting, ICCV Workshop, 2011.

 - Practical Spectral Characterization of Trichromatic Cameras, TOG, 2011.

Other minor corrections:

(line 51) C_i -> C_c

(line 51) often i -> often c

(above the Eq.(2)) R_{i,lambda} -> R_{c,lambda}

(line 53) R_i

(line 89) estimationg -> estimation

(line 130) known known -> known

(line 156) n:th -> n-th

(line 246) Both constant grating and monochromatic only, constant grating -> It was hard to understand what methods the authors are mentioning.

Author Response

1) I could not fully understand what the numerical values evaluated in Table I are. Are they RGB value errors for color patches? Please clarify. If I assume that Table I evaluates the RGB errors, the numerical evaluation for camera spectral response estimation errors (in the wavelength domain) should be added to directly evaluate the accuracy improvement of the estimated camera spectral functions by using the proposed method.

Table 1 does indeed evaluate RGB value errors for color patches. This has been clarified in the caption and the table headers. We also added a new table (Table 2) that evaluates the spectral responses as suggested, comparing the different methods against the ground truth available for the 400-700nm range. The proposed method achieves the lowest mean error also on this metric. 

2) Since all the compared methods use diffraction images, it still remains unclear how much the estimation accuracy is improved by exploiting the diffraction grating, in comparison with a standard method using only color patches. To confirm the effectiveness of using the diffraction grating, the comparison with the baseline method using color patch data only (with known illumination spectra and a linear basis model for the camera responses, e.g., [10]) should be added. (This method may correspond to "color patch only" in the author's method.)

We now provide an additional comparison against a method using only color patches. The method is described in Section 4.2.4, and the results are shown in Tables 1 and 2. The method does not exactly correspond to any published method, but is instead a variant of the proposed method that omits the diffraction images but regularizes for smoothness of spectral responses similar to [24]. This new baseline achieves good RGB error, but as shown in Table 2 performs very poorly in terms of the spectral response, demonstrating the value of the diffraction images.

[24] Practical Spectral Characterization of Trichromatic Cameras, TOG, 2011.

3) Also, it is hard to judge and compare the estimation accuracy from Figure 7. Each estimation result should be overlaid with the ground truth.

Ground truth spectral responses have now been overlaid in each of the result plots in Figure 7.

4) Throughout the paper, the authors use the term “color filter” to represent an optional color filter placed in front of the camera system. However, it is a little confusing because “color filter” is also commonly used to represent on-sensor RGB filters to form RGB responses (i.e., R_c). Thus, a more careful explanation is necessary for the usage of “color filter” in the paper.

The term "color filter" refers in our work to a transmissive filter that selectively attenuates some portions of the spectra to which the camera sensor is sensitive to, placed in front of or immediately behind the objective lens, and can be understood as the Bayer color filter integrated onto the camera sensor. This has been clarified in Section 2 (Previous methods).

5) In addition to the above comment, the explanation “the color patch images are captured using a set of color filters” in line 159 is also a little confusing because the color patch is usually considered as a reflective patch (e.g., X-Rite ColorChecker). Thus, a more careful explanation is recommended. Also, Figure 5 should be cited in this subsection.

Figure 5 has now been cited in the proper section. It is true that most commonly color patches are reflective instead of transmissive filters. We have used transmissive filters because the spectrum of the emitted light through such transmissive filters can be measured using an spectral illuminance meter. Section 3.3 (Color patch images) now clarifies that transmissive color filters are being used. 

6) The explanation of the optimization cost and procedure in section 3.5 should be improved.

 - In line 198-199, there is the description that the grating efficiency is modeled by a quadratic polynomial. However, there is no mathematical explanation for the specific form of the used model. Thus, optimization parameters for the grating efficiency remain unclear.

- For a series of equations on page 8, it could be clearer that the authors first show the final optimization cost “L_total” and then explain each term by introducing the detailed meaning of mathematical symbols.

- Please clarify the optimization parameters to be estimated to minimize “L_total”.

The grating efficiency model has now been added in Section 3.5 (Parameter estimation), and we now list the set of parameters that are being optimized in the same section. We also changed the order of the loss equations to improve clarity, and explain the different components of the total loss more clearly.

7) Two related papers are recommended to include.

 - Quick Approximation of Camera’s Spectral Response from Casual Lighting, ICCV Workshop, 2011.

 - Practical Spectral Characterization of Trichromatic Cameras, TOG, 2011.

We agree these are relevant papers, and now discuss them in Section 2 (Previous Methods) and Section 4.2.4 (Color patches only).

Other minor corrections:

 (line 51) C_i -> C_c

(line 51) often i -> often c

(above the Eq.(2)) R_{i,lambda} -> R_{c,lambda}

(line 53) R_i

(line 89) estimationg -> estimation

(line 130) known known -> known

(line 156) n:th -> n-th

(line 246) Both constant grating and monochromatic only, constant grating -> It was hard to understand what methods the authors are mentioning.

We have corrected all of these.

Reviewer 2 Report

I consider the revised article could be an interesting work with only one major drawback: the comparisons presented do not include other state of the art  methods or the explanation of the comparisons is ambiguous.  Authors cite articles that seem to be used as reference but finally it seems that it is not done (or is ambiguous).

In any case this comparison should be added, even if the results were not as good as using much more expensive equipment.

On the other hand the article must be thoroughly proofread (for example the incorrect use of "alone" and many others).

Other concerns:

Figure 5 is not referenced in the text.

Use of i and c (line 51 and Equation 1)

Define “lambda” before the first use

include a image of the actual device, not a diagram in Figure 1 (or both).

Explain figures in text not in captions.

Respect “LED producing white light”, what degrees kelvin is the white light emitted? Is it important?

“Uniform distribution” is random uniform distribution?

Could an economic cost analysis be interesting?  (only a proposal)

Author Response

I consider the revised article could be an interesting work with only one major drawback: the comparisons presented do not include other state of the art  methods or the explanation of the comparisons is ambiguous.  Authors cite articles that seem to be used as reference but finally it seems that it is not done (or is ambiguous).

In any case this comparison should be added, even if the results were not as good as using much more expensive equipment.

We added two new comparisons in the revised version, against a method using only color patch images and against ground truth data measured using a monochromator. The results of these comparisons are presented in Tables 1 and 2; the latter is a new table added in response to comments by Reviewer 1.

The comparison method using only color patches (described in Section 4.2.4) demonstrates the importance of the diffraction images. It does not correspond exactly to any published method since they rely on reflective color charts, whereas we used transmissive color filters and hence the exact algorithms are not applicable. However, it uses similar regularization to encourage smooth spectral responses, and is optimized in the same way as our proposed approach to minimize differences caused by other reasons than use of the diffraction images.

We have also now directly compared our results to ground truth spectral responses gathered using a monochromator, as explained in Section 4. The measurements were made by Kawakami et al. [1] and Jiang et al. [10] for the same camera models as used in our experiments, made publicly available in [28,29].

On the other hand the article must be thoroughly proofread (for example the incorrect use of "alone" and many others).

The manuscript has been thoroughly proofread and checked for grammar and spelling. The manuscript now consistently follows US English spelling (e.g. regularise -> regularize, analyse -> analyze, etc.).

Other concerns:

Figure 5 is not referenced in the text.

Figure 5 has now been referenced in the text in section 3.3 (Color patch images).

Use of i and c (line 51 and Equation 1)

The use of i and c indices has now been corrected and is used consistently.

Define “lambda” before the first use

Lambda is now defined (in Section 2, Previous Methods) to refer to wavelength.

include a image of the actual device, not a diagram in Figure 1 (or both).

We complemented Figure 1 so that it now includes both a schematic diagram and a photograph of the actual device.

Explain figures in text not in captions.

We shortened captions for Figures 2-5, moving the details for the main text. However, we decided to keep the description for Figure 1 in the caption, since it refers directly to specific elements in the illustration. We also removed some unnecessary replication with the main text in Figure 7 presenting the main empirical results.

Respect “LED producing white light”, what degrees kelvin is the white light emitted? Is it important?

The correlated color temperature of the white light used in the production of color patches was 10833 K and is now mentioned also in the manuscript. The CCT of the white light is not as important as the overall uniformity of the spectral distribution over the whole measurement range (i.e. at least 380-720nm). Areas of low intensity result in less uncertain estimates, as explained in Section 4.2.5; the highest uncertainty is in the blue end of the spectrum, for which both the halogen and LED lights have lower intensity.

“Uniform distribution” is random uniform distribution?

Uniform distribution has been clarified to refer to the random uniform distribution. 

Could an economic cost analysis be interesting?  (only a proposal)

We originally omitted the cost analysis since detailed analysis of the components felt out of place in a scientific publication. We now mention in Section 1 (Introduction) that the overall material costs of the device are below 100 euros and that it can be assembled in a few hours by a skilled person, also clarifying that the method requires use of spectral illuminance meter during construction (but not after that). For your interest, we provide below a more detailed breakdown of the costs, but are not including it in the manuscript. We are also planning on writing a separate non-scientific blog post with more detailed construction instructions once the paper is published.

The enclosures for the equipment used to capture diffraction photos and color patch photos were manufactured out of plywood by laser cutting and 3D printing. The material cost for the enclosure is estimated to be 5 euros for the plywood and 5 euros for the PLA plastic used for 3D printing, so a total of 10 euros. The unit costs of the LEDs and transmissive color filters used are all very low, in the order of at most tens of euro cents each: 16 filters and 11 LEDs, so approximately 5 euros accounting for wires and supplies. The cost of the PCB is approximately 1 euro in low quantities, and a 5V power source for the LED can be bought for 5 euros.

A large piece of plastic diffraction grating can be bought for approximately 10 euros, but only a small piece is needed for the equipment. Frames for the slit and diffraction grating were laser cut out of acrylic plastic, for which the estimated cost is 4 euros. The diffraction apparatus was mounted on a polyurethane (PU) sheet that provided a level plane that is also light weight. The PU sheet could be replaced with any other level and a black table would suffice for the purpose. A halogen light is also needed as a light source. Halogen light bulbs maybe difficult to buy in some locations, so an LED with a broad and even spectral distribution can be used instead. The estimated cost for the light source is 10-20 euros. 

The single most expensive equipment is a device for measuring spectra of light. We used a Hopoocolor OHSP-350C spectral illuminance meter, which can probably be bought for 1000-1200 euros. However, the spectra of the light sources only needs to be measured once, and hence a borrowed or rented device is sufficient.

Not including the spectral illuminance meter, the total cost of materials and machine operating costs is hence approximately 60-70 euros. The equipment can be assembled in approximately 3 hours by a reasonably skilled person, which means that in practice the labor cost is likely to override the material expenses.

Round 2

Reviewer 1 Report

I confirmed that the authors have adequately addressed my review comments in the revised manuscript.

Reviewer 2 Report

The authors have correctly addressed the main concern of my first review, as well as all the minor revisions proposed.